# Expression Profile of microRNAs during Development of the Hypopharyngeal Gland in Honey Bee, *Apis mellifera*

**DOI:** 10.3390/ijms232112970

**Published:** 2022-10-26

**Authors:** Kaixin Qin, Fuping Cheng, Luxia Pan, Zilong Wang

**Affiliations:** 1Honeybee Research Institute, Jiangxi Agricultural University, Nanchang 330045, China; 2Jiangxi Province Key Laboratory of Honeybee Biology and Beekeeping, Nanchang 330045, China

**Keywords:** *Apis mellifera*, hypopharyngeal gland, miRNA, expression profile, miRNA target

## Abstract

The hypopharyngeal gland is an important organ for honey bees to secrete royal jelly, and its secretory activity varies with the age of workers. However, by now, the regulation mechanism of hypopharyngeal gland development is still unclear. Here, the expression profiles of miRNAs in the hypopharyngeal gland of newly emerged workers, nurses, and foragers were investigated via small RNA sequencing. From these three stages, 81 known miRNAs and 135 novel miRNAs have been identified. A total of 85 miRNAs showed expression differences between different development stages, and their target genes were predicted to range from 1 to more than 10. Many of the differentially expressed miRNAs and target genes are related to growth and development or apoptosis. Moreover, dual-luciferase-reporter assays verified that novel-miR-11 directly targets the 3′-untranslated regions of LOC410685 (inactive tyrosine-protein kinase 7) and LOC725318 (uncharacterized protein). These results suggested that miRNAs were widely involved in the developmental regulation of the hypopharyngeal gland in honey bees.

## 1. Introduction

The hypopharyngeal gland is a pair of grape-like glands located in the head of honey bees. One of its terminals opens at the lateral corner of the oral slice, and the other end is free. Each hypopharyngeal gland consists of a main secretory tube and hundreds of hypopharyngeal acini. The hypopharyngeal gland of nurses is highly developed to synthesize and secrete royal jelly proteins for feeding larvae and queens.

The development of the hypopharyngeal gland is age-dependent [1]. It begins to differentiate in the pupae stage. The hypopharyngeal acini of the newly emerged workers (NEW) were flat, pale white, not hypertrophied, and without secretory activity. When the workers are 6–12 days old, the hypopharyngeal gland reaches its maximum size, and the acini are in the most hypertrophic state with the highest secretory activity. When the workers become foragers, the hypopharyngeal glands shrink to the smallest size, lose their secretory activity, and turn pale yellow [2,3,4].

The hypopharyngeal glands are highly developmentally plastic, which is influenced by many factors, such as the existence of a queen, season, larval signals, nutrition, hormones [5], pheromones [6,7], and diseases. When the number of larvae in the colony increases and the feeding capacity of the colony is insufficient, the hypopharyngeal acini of the foragers will develop again and restore their secretory activity [8]. This change also occurs in some workers when the queen is lost [8].

The molecular mechanism of the hypopharyngeal gland development in honey bees has been studied extensively. Many differentially expressed genes (DEG) between different developmental stages of hypopharyngeal glands in *A. mellifera* workers were identified through RNA-Seq [9] or digital gene expression profiling analysis [10]. Moreover, proteome analysis showed that the protein expression pattern of the hypopharyngeal gland changed with the development of the workers and the transformation of their tasks [11]. Moreover, the study of epigenetic modification in the hypopharyngeal glands of workers at inactive winter stages and arouse stages in spring indicated that DNA methylation is involved in the activation of hypopharyngeal glands in overwintering workers [12].

microRNAs are a kind of endogenous non-coding single-stranded RNA molecule with a length of 18–24 nt existing widely in various eukaryotic organisms. They degrade the mRNA of the target genes or effectively inhibit the translation of the target genes mainly through incomplete matching with the 3′-untranslated regions (3′UTR) of the target gene’s mRNA [13]. They play an important role in the process of cell formation [14], differentiation [15,16], apoptosis [17,18], embryonic development [19,20], neurogenesis [21,22], immune response [23,24] and disease occurrence [25,26]. By now, many miRNAs have been identified in *A. mellifera* through different methods [27,28]. Functional studies of miRNAs have shown that miRNAs in honey bees are involved in regulating biological processes such as caste differentiation [29], division of labor [30,31], learning, and memory [32].

In this study, small RNA sequencing was adopted to identify differentially expressed miRNAs (DEmiRNAs) between three developmental stages of the *A. mellifera* hypopharyngeal gland. Finally, we identified many DEmiRNAs between these three stages, and many of them are related to development or apoptosis. The results will provide important information for understanding the regulation mechanism of hypopharyngeal gland development and synthesis and secretion of royal jelly proteins and lay the foundation for further research on the function of miRNAs in regulating the development of the hypopharyngeal gland in honey bees.

## 2. Results

### 2.1. Statistics of Sequencing Results

To investigate the effects of miRNAs on the development of the *A. mellifera* hypopharyngeal gland, we constructed and sequenced nine sRNA libraries from the hypopharyngeal gland at three different development stages. The original reads of nine samples obtained by sequencing range from 12,815,475 to 50,870,653 (Table 1). After removing low-quality sequences, adapter sequences, and sequences less than 18 nt, the remaining sequences ranged from 10,560,784 to 44,999,092. In all nine samples, small RNAs were distributed between 18 and 30 nt and concentrated in 22 nt (Appendix A). The Q30 of all the samples was higher than 96.96%. After removing ncRNAs and repetitive sequences, the remaining unannotated reads ranged from 4,285,110 to 19,548,174, and about 39.89% to 54.64% of unannotated reads were mapped to the reference genome. The person correlations between biological replicates were above 0.89 (Appendix A), suggesting the reliability of the sRNA-seq result.

### 2.2. Known miRNAs and Novel miRNAs

By comparing all clean reads with the precursors and maturities of miRNAs from *A. mellifera* in the miRNA database (miRBase v22), a total of 81 known miRNAs were identified (Table 2). Novel miRNAs were predicted based on the typical stem-loop hairpin structures of the miRNA precursors. From these nine samples, 135 novel miRNAs were predicted using miRDeep2 software (Table 2). Of the 216 miRNAs, 170 miRNAs were shared by the three kinds of bees, 13 miRNAs were unique to NEW, and 1 miRNA was unique to forager (Figure 1A). Many miRNAs belong to conserved miRNA families (Appendix A).

Seven novel miRNAs were selected for validation of their existence by RT-PCR. The results showed that all seven novel miRNAs were detected in the hypopharyngeal gland of NEWs, nurses, and foragers (Figure 1B).

### 2.3. Differentially Expressed miRNAs

A total of 85 miRNAs were differentially expressed at different developmental stages of the hypopharyngeal gland in *A. mellifera*. There were 68 DEmiRNAs in NEW vs. nurse, including 31 up-regulated and 37 down-regulated miRNAs in NEW, and 15 DEmiRNAs were detected in nurse vs. forager, including 7 up-regulated and 8 down-regulated miRNAs in nurse. In the NEW vs. forager, 72 DEmiRNAs were identified, including 34 up-regulated miRNAs and 38 down-regulated miRNAs in NEW. By Venn diagram analysis, 5 DEmiRNAs, including ame-miR-315-5p, novel-miR-104, novel-miR-5, novel-miR-79, and novel-miR-23, were shared in these three comparisons (Figure 2A, Appendix A). Expression clustering analysis showed that these DEmiRNAs were divided into two major clusters according to their expression patterns (Figure 2B).

Nine DEmiRNAs in NEW vs. nurse, six DEmiRNAs in nurse vs. forager, and eight DEmiRNAs in NEW vs. forager were randomly selected for qPCR validation of the sequencing results. It showed that the expression of all these DEmiRNAs had the same expression trend as the sRNA-seq results, indicating the reliability of the sequencing results (Figure 3).

### 2.4. Target Genes of DEmiRNAs

The target genes of all DEmiRNAs were predicted by miRanda and targetscan, but only 54 DEmiRNAs have successfully predicted target genes. To obtain annotation information on the target genes, their protein sequences were blasted against the NR, Swiss-Prot, GO, COG, KEGG, KOG, and Pfam databases. In total, 862 DEmiRNA target genes were predicted in the comparisons NEW vs. nurse, nurse vs. forager, and NEW vs. forager (Appendix A).

GO classification analysis showed that all the target genes of DEmiRNAs in the three comparisons were significantly enriched in 11, 5, and 17 GO terms, respectively (*p* < 0.05) (Appendix A). Of them, many GO terms are related to development or protein synthesis, such as “cellular differentiation”, “Notch signaling pathway”, “regulation of developmental process”, “protein homooligomerization”, “translational initiation”, and “translation initiation factor activity”.

KEGG pathway analysis of the target genes of DEmiRNAs showed that in NEW vs. nurse, 47 pathways were enriched, and the top enriched was the hippo signaling pathway-fly (Figure 4A, Appendix A). In nurse vs. forager, 5 pathways were enriched (Figure 4B, Appendix A). In NEW vs. forager, 52 pathways were enriched, among which the highest enriched pathway was neuroactive ligand-receptor interaction (Figure 4C, Appendix A).

### 2.5. miR-11 Targets LOC410685 and LOC725318

Considering that from NEW to nurse is a critical period for the development of the hypopharyngeal gland, we focused on DEmiRNAs in NEW vs. nurse. We noticed that some DEmiRNAs were highly expressed in these two stages. We predicted the binding sites of novel-miR-11 on its 25 target genes and found those of LOC410685 and LOC725318 lay on 3′UTRs. Thus, we selected these two target genes for experimental verification.

To test whether novel-miR-11 directly targets 3′UTR of LOC410685 and LOC725318 gene, we used dual-luciferase-reporter assays to verify the prediction results. The results showed that the luciferase ratio was significantly reduced in LOC410685-WT + novel-miR-11 and LOC725318-WT + novel-miR-11 compared with the three controls (Figure 5), indicating that novel-miR-11 actually targets 3′UTR of LOC410685 and LOC725318.

## 3. Discussion

MiRNAs are an important class of small RNAs in organisms, which participate in the regulation of many biological processes through inducing gene silencing, such as cell growth, development, gene transcription, and translation. Using small RNA sequencing technology, 85 DEmiRNAs were identified between different developmental stages of the honey bee hypopharyngeal gland, which revealed that miRNAs are involved in the developmental regulation of the honey bee hypopharyngeal gland.

In NEW vs. nurse, several miRNAs were related to growth and development, including miR-263a, miR-317, and miR-34. In *Drosophila*, miR-263a/b projects *Drosophila* mechanosensory bristles by down-regulating the pro-apoptotic genes [33], suggesting that miR-263a/b may have a developmental regulation role in mechanosensory bristles. Moreover, in the posterior silk gland of fifth-instar *Bombyx mori*, miR-263a has a very high expression level and was considered to be involved in the posterior gland development [34]. MiR-317 has been shown to play an important role in regulating *Drosophila* larval ovary morphogenesis [35]. In *Bactrocera dorsalis*, injecting antago-317 reduced the pupation rate and shortened pupation time [36], suggesting that miR-317 regulates the development of *Bactrocera dorsalis* during the pupal stage. MiR-34 is one of the most conserved miRNAs from worms to humans and plays an important role in insect segmentation [37]. During the early development of the honey bee brain, miR-34 causes differences in brain development between the queen and worker by down-regulating the expression of hexamerin [38]. The differential expression of these miRNAs between NEW and nurse suggests that they may be involved in the developmental regulation of the hypopharyngeal gland at the early stage of adult bees.

During the nursing stage, the hypopharyngeal glands of nurses secrete a large amount of royal jelly that is rich in proteins. We found that some target genes of ame-miR-263a-5p, novel-miR-124, and ame-miR-3785-3p are associated with posttranslational modifications (Appendix A). Moreover, these miRNAs were down-regulated in nurses, indicating a higher expression of their target genes in the nursing stage. Therefore, these miRNAs may affect the posttranslational modification of proteins expressed in the hypopharyngeal gland.

KEGG enrichment analysis of the targets of DEmiRNAs in NEW vs. nurse showed that the hippo signaling pathway was significantly enriched (Appendix A). Hippo signaling was originally identified in *Drosophila* and functions to control organ growth and maintain tissue homeostasis [39,40]. In this pathway, we noticed two target genes, crumbs and hemicentin, which are involved in the development of tissues. Crumbs was identified as a regulator of apicobasal polarity for promoting the development of the embryonic ectoderm epidermal cells [41]. It also acts as a growth regulator to regulate cell proliferation during *Drosophila* eye development [42]. Hemicentin is a highly conformed extracellular matrix protein that plays an important role in the development and homeostasis of tissues [43]. Therefore, DEmiRNAs associated with this pathway may regulate cell proliferation and organ growth of the honey bee hypopharyngeal gland.

Compared with nurses, the hypopharyngeal glands of foragers undergo atrophy due to apoptosis [2]. We found that two DEmiRNAs in nurse vs. forager, miR-210 and miR-29, are related to apoptosis (Appendix A). miR-210 regulates the target gene SIN3A in human lung cancer to participate in proliferation and apoptosis in non-small cell lung cancer (NSCLC) [44]. MiR-29 can induce apoptosis in acute myelogenous leukemia (AML) cell lines and primary samples [45]. Therefore, miR-210 and miR-29 may be related to the atrophy of the hypopharyngeal gland caused by cell apoptosis in foragers.

Through dual-luciferase-reporter assays, we found that novel-miR-11 directly targets the 3′UTR of LOC410685 and LOC725318. LOC410685 encodes tyrosine-protein kinase 7 (PTK7), which is associated with the Wnt signaling or Wnt/planar cell polarity pathway in mouse [46] and *Xenopus* embryos [46,47], and PTK7-deficient cause developmental defects [48]. LOC725318 codes for a transmembrane protein without a defined function. The dual-luciferase assay showed that novel-miR-11 decreased the expression of LOC410685 and LOC725318, indicating that novel-miR-11 directly regulates LOC410685 and LOC725318. Therefore, novel-miR-11 may regulate the development of the hypopharyngeal gland by targeting LOC410685 and LOC725318.

In summary, our study indicates that miRNAs are widely involved in regulating hypopharyngeal gland development or royal jelly secretion and provides a basis for further research on the role of these candidate miRNAs in the development of the hypopharyngeal gland of *A. mellifera*.

## 4. Materials and Methods

### 4.1. Insects

The *A. mellifera* colonies used in this study were raised according to standard technology by Honeybee Research Institute at Jiangxi Agricultural University. Three colonies with similar populations (each containing 10 combs and about 30,000 bees) were used in this experiment. A comb with many worker pupa was taken out from each colony and placed in an incubator at 34 °C and 80% humidity [49]. After the emergence of the adult workers, 50 newly emerged workers from each colony were chosen for dissecting hypopharyngeal glands. The remaining bees were marked on the back of the thorax with colors and put back into the original hive. The nurses (continuously inserting their heads into the nest with small larvae) and foragers (pollen harrows on the hind feet) were randomly caught from each group. The hypopharyngeal glands were dissected in saline solution. All hypopharyngeal gland samples were ground into powder and mixed with TransZol (TransGen Biotech, Beijing, China) in a plastic Eppendorf tube. Then, all samples were promptly frozen and stored in liquid nitrogen until RNA extraction.

### 4.2. Construction of Small RNA Libraries and High Throughput Sequencing

Total RNAs were extracted from the nine hypopharyngeal gland samples. The concentration and integrity of the total RNAs were measured by NanoDrop 2000 (Thermo Fisher Scientific, Wilmington, DE, USA) and Agilent Bioanalyzer 2100 system (Agilent Technologies, Santa Clara, CA, USA), respectively. Then, small RNA libraries were prepared with VAHTSTM Small RNA Library Prep Kit for Illumina^®^ (Vazyme, Nanjing, China). At first, the 3′ adapter and 5′ adapter were added to the two ends of RNA molecules in turn. Subsequently, these RNA fragments with adapters were reverse transcribed into cDNA and amplified by PCR. The amplification products were purified with VAHTSTM DNA Clean Beads (Vazyme, Nanjing, China) and 6% TBE gel and then unchained. The single-chain molecules were added to Illumina sequencing chips (flowcell) and were sequenced using Illumina NovaSeq 6000 sequencer. Small RNA fragments of 18–30 nt were isolated from total RNA and purified with 15% TBE-urea gel (Invitrogen Company, Carlsbad, CA, USA).

The original image data obtained by sequencing was converted into sequence data by base calling. After removing low-quality reads, contamination reads, and reads shorter than 18 nt or longer than 30 nt, clean reads were obtained. 

### 4.3. Identification of Known miRNAs and Novel miRNAs

Using Bowtie (v1.0.0) [50] software, all the clean reads were compared with Silva, GtRNAdb, Rfam, and Repbase databases for sequence comparison, and ncRNAs such as ribosomal RNA (rRNA), transfer RNA (tRNA), intranuclear small RNA (snRNA), nucleolar small RNA (snoRNA) and repetitive sequences were filtered out to obtain unannotated reads. Then, the distribution and expression of unannotated reads in the genome version 4.5 of *A. mellifera* were analyzed using Bowtie software. After that, the unannotated reads were compared with the precursors and maturities of *A. mellifera* miRNAs in the miRNA database (miRBase v22), and the numbers of known miRNAs in the samples were obtained. 

Reads without known miRNAs identified were used to identify novel miRNAs based on the typical structural features of miRNA precursors. The novel candidate miRNAs were identified using miRDeep2 (v2.0.5) [51] software by intercepting the genomic sequences of *A. mellifera* that small RNAs matched and analyzing their secondary structures, digestion sites of the Dicer enzyme, and energy.

### 4.4. Screening of Differentially Expressed miRNAs

All the miRNAs expressed in each sample were counted to determine their expression levels. Then, the expression level of each miRNA was normalized using the TPM algorithm [52]. After normalization, DEmiRNAs between two development stages were screened using the DESeq2 R package [53] with the criteria |log2(FC)| ≥ 1.00 and FDR ≤ 0.05.

### 4.5. Prediction of miRNA Targets

MiRNA targets were predicted by two kinds of predictive software, including miRanda [54] and Targetscan [55], and target genes with higher scores in the analysis of the two software were selected as the final targets. The predicted target genes were annotated by aligning them with the NR, Swiss-Prot, GO, COG, KEGG, KOG, and Pfam databases using the BLAST program. GO and KEGG enrichment analysis of the predicted target genes were performed in the Gene Ontology (GO) database and the Kyoto Encyclopedia of Genes and Genomes (KEGG) database, respectively.

### 4.6. Verification of Novel miRNAs and DEmiRNAs

According to the above-mentioned methods, the hypopharyngeal gland of NEW, nurses, and foragers was collected. The total RNA of each sample was extracted using TransZol Up Plus RNA Kit (TranzGen Biotech, Beijing, China) and was reverse transcribed using Mir-X miRNA First-Strand Synthesis Kit (TAKARA, Beijing, China). Verification of novel miRNAs was performed by stem-loop RT-PCR. Total RNAs were reverse transcribed using RevertAid First-Strand cDNA Synthesis Kit (Thermo Fisher Scientific, Waltham, MA, USA) and stem-loop primers. RT-PCR was performed by T100 thermo cycler (BIO-RAD, Hercules, CA, USA) under the following: pre-denaturation step at 95 °C for 5 min; 35 amplification cycles of denaturation at 95 °C for 30 s, 55 °C for 45 s, 72 °C for 45 s and elongation at 72 °C for 10 min. The PCR products were detected on 1% agarose gel electrophoresis.

DEmiRNAs were verified by RT-qPCR. The RT-qPCR primers were designed using miRprimer2 [56], and U6 was used as a reference gene (Appendix A). RT-qPCR was performed by QuantStudio5 (Thermo Fisher Scientific, Waltham, MA, USA) in 10 μL reaction volume containing 5 μL TB Green Premix Ex Taq Ⅱ (TAKARA, Beijing, China), 0.4 μL forward and reverse primers, 2 μL cDNA, and 2 μL double-distilled H_2_O. The relative expression levels of DEmiRNAs were analyzed using 2^−ΔΔCT^ methods, and “*t*-Test” was used for statistical analysis by SPSS software.

### 4.7. Dual-Luciferase-Reporter Assay

The psiCHECK2 vector (Promega, Madison, WI, USA) that contains the coding region of Firefly luciferase and Renilla luciferase was used as a basic vector to perform the luciferase-reporter assays. Firefly luciferase was used as an internal control. The 3′UTR fragments containing the predicted novel-miR-11 target sites of the target genes and their corresponding mutant sequences were synthesized by General Biology Company and were inserted into the psiCHECK2 vector using Xho-I and Not-I restriction endonucleases (Appendix A). Then, the recombinant vector and novel-miR-11 (or miR-NC: UCACAACCUCCUAGAAAGAGUAGA) were co-transfected into 239T cells, respectively. For each target gene, four experimental conditions were tested: target-WT + miR-NC, target-WT + novel-miR-11, target-MUT + miR-NC, and target-MUT + novel-miR-11. Forty-eight hours after transfection, Firefly and Renilla luciferase activities were measured by Dual-Luciferase^®^ Reporter Assay System (Promega, Madison, WI, USA) using Lux-T020 High-Sensitivity Tube Luminescence Detector (BLT, Guangzhou, China). The ratios of two luciferases (Ren/FF) were calculated, and the “*t*-Test” was used for statistical analysis by SPSS software.

## Figures and Tables

**Figure 1 ijms-23-12970-f001:**
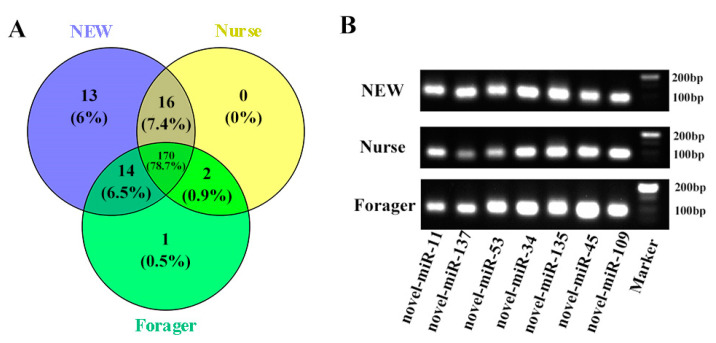
Identification of miRNAs. (**A**) Venn diagram of all miRNAs in three stages. (**B**) Stem-loop RT-PCR confirmation of novel miRNAs in hypopharyngeal glands of three stages.

**Figure 2 ijms-23-12970-f002:**
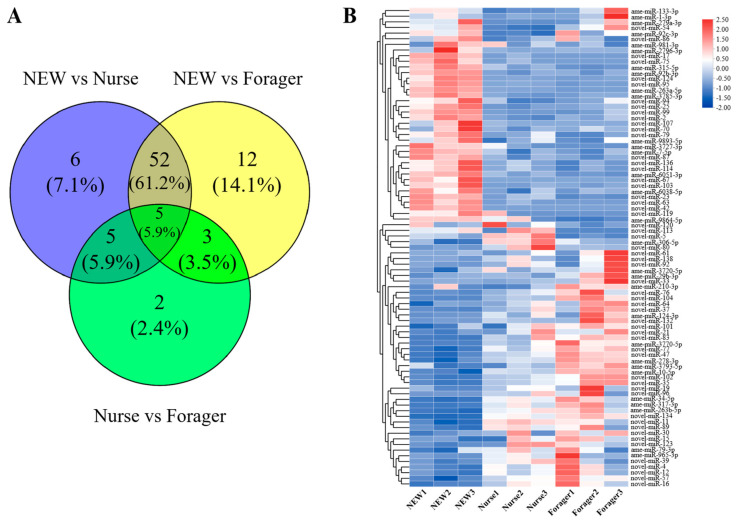
Differentially expressed miRNAs. (**A**) Venn diagram of DEmiRNAs in three comparisons. (**B**) Expression clustering of miRNAs in nine samples. Red indicates high expression, while blue indicates low expression.

**Figure 3 ijms-23-12970-f003:**
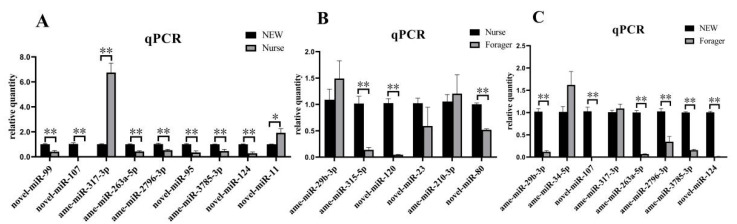
Validation of DEmiRNAs by qPCR. Data are expressed as mean ± standard error of mean (SEM). * means *p* < 0.05, and ** means *p* < 0.01. (**A**) qPCR validation of DEmiRNAs in NEW vs. nurse. (**B**) qPCR validation of DEmiRNAs in nurse vs. forager. (**C**) qPCR validation of DEmiRNAs in NEW vs. forager.

**Figure 4 ijms-23-12970-f004:**
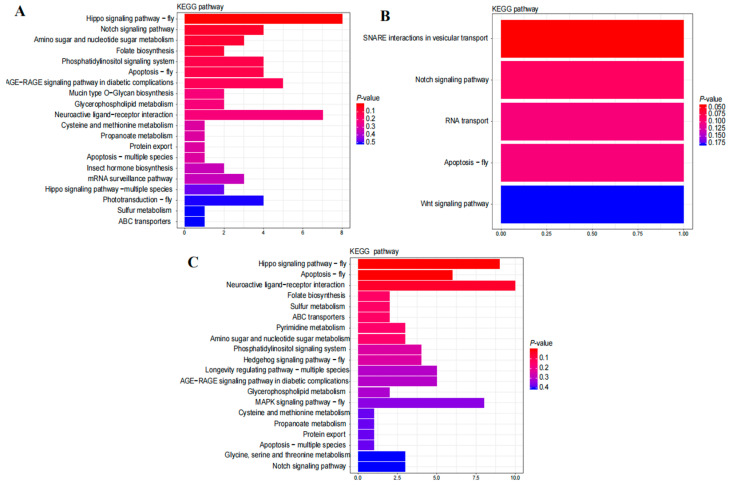
KEGG pathway enrichment of DEmiRNA target genes. Only the top 20 pathways were presented in (**A**,**C**). (**A**) Enriched pathways in NEW vs. nurse. (**B**) Enriched pathways in nurse vs. forager. (**C**) Enriched pathways in NEW vs. forager.

**Figure 5 ijms-23-12970-f005:**
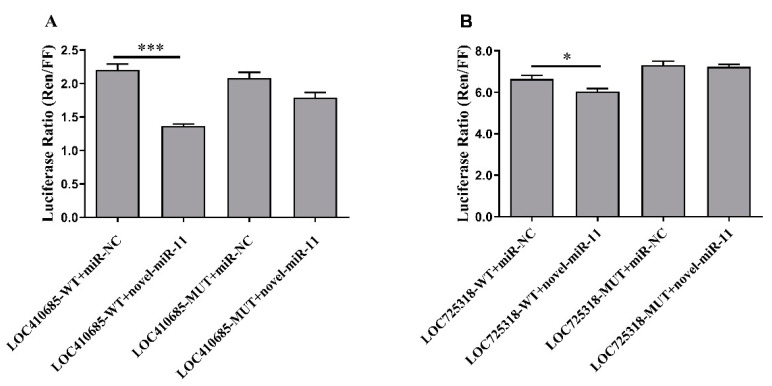
Dual-luciferase-reporter assays of novel-miR-11 target interactions. * means *p* < 0.05, and *** means *p* < 0.001. (**A**) The interaction between novel-miR-11 and LOC410685. (**B**) The interaction between novel-miR-11 and LOC725318.

**Table 1 ijms-23-12970-t001:** Statistics of sequencing results.

Sample	Raw Reads	Clean Reads	Q30 (%)	Unannotated Reads	Mapped Unannotated Reads
NEW1	14,566,809	10,560,784	98.14	4,495,092	2,299,248 (51.15%)
NEW2	50,870,653	44,999,092	96.96	19,548,174	10,379,450 (53.10%)
NEW3	25,761,918	19,506,666	97.15	8,216,128	4,194,771(51.06%)
nurse1	17,193,017	11,653,112	97.56	5,655,312	2,329,792 (41.20%)
nurse2	24,858,856	20,396,246	97.34	8,819,202	4,017,379 (45.55%)
nurse3	13,970,243	11,579,964	98.18	4,423,128	2,260,265 (51.10%)
forager1	12,815,475	10,594,494	98.05	4,722,875	1,884,181 (39,89%)
forager2	15,885,906	12,159,719	98.17	5,701,605	2,527,884 (44.34%)
forager3	14,744,683	12,205,485	98.27	4,285,110	2,341,378 (54.64%)

**Table 2 ijms-23-12970-t002:** Summary of miRNAs in nine samples.

Sample	Known miRNAs	Novel miRNAs	Total
NEW	79	134	213
Nurse	68	120	188
Forager	69	118	187
Total	81	135	216

## Data Availability

All the raw sequencing reads for all the samples have been submitted to the National Centre for Biotechnology Information (NCBI) sequence read archive and deposited under the BioProject number: PRJNA872281.

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
