# Peer review of "Expression Profile of microRNAs during Development of the Hypopharyngeal Gland in Honey Bee, Apis mellifera"

_ijms, 2022, doi:10.3390/ijms232112970_

Round 1

Reviewer 1 Report

 Review of “Expression profile of microRNAs during development of the hypopharyngeal gland in honey bee, Apis mellifera” by Qin et al.

The authors investigate microRNA expression at different developmental stages of the honeybee Apis mellifera.  They identify many new and known miRNAs.  They identify differentially expressed miRNAs between life stages and also attempt to determine what types of genes the miRNAs might control.

Overall, I found this to be a reasonable study of miRNAs.  The scope of the study is somewhat small, given the sample size of n = 9.  But I did like the functional reporter assays, and thought that these strengthened the manuscript.  And the research team does identify some new putative miRNAs.  I have only a few suggestions.

 The authors state that “only 54 DEmiRNAs have successfully predicted target genes”.  But this is a bit worrying.  Is this normal for this type of study?  If these are real miRNAs why is the percentage of putative targets so low? 

 The suggestion of finding 135 new miRNAs is intriguing.  Do these ‘new’ miRNAs show any homology with miRNAs from more distantly related taxa at all?

 Can the authors provide a bit more information on exactly where in the genome these miRNAs are being transcribed?  That is, where are these miRNA genes located?  Which chromosomes?  Are the novel miRNAs being expressed from certain clusters?  I’d be curious to know more information.

 I was impressed that the research team did some analyses to see if the putative miRNA actually targeted a particular mRNA.  They seem to have found results suggesting that it did.  I would have welcomed more directed functional assays. 

Author Response

Dear editor Codruta Cormos,

We greatly appreciate you and two reviewers giving so many very helpful and useful comments for our manuscript “Expression profile of microRNAs during development of the hypopharyngeal gland in honey bee, Apis mellifera” (ijms-1950892). These comments greatly improve the quality of our manuscript.

We have carefully considered each point raised in the reviewers, and provide point-by-point responses to the reviewers’ comments below. The corrections are highlighted in red in the revised version. We hope we have addressed all comments, and please let us know if more information is needed. Thank you once again for considering our manuscript.

Best wishes,

Prof. Dr. Zilong Wang

Director of Honeybee Research Institute

Jiangxi Agricultural University

Nanchang, Jiangxi, 330045

  1. R. of China

Response to Reviewer 1 Comments
Comments:

Review of “Expression profile of microRNAs during development of the hypopharyngeal gland in honey bee, Apis mellifera” by Qin et al.

The authors investigate microRNA expression at different developmental stages of the honeybee Apis mellifera. They identify many new and known miRNAs. They identify differentially expressed miRNAs between life stages and also attempt to determine what types of genes the miRNAs might control.

Overall, I found this to be a reasonable study of miRNAs. The scope of the study is somewhat small, given the sample size of n = 9. But I did like the functional reporter assays, and thought that these strengthened the manuscript. And the research team does identify some new putative miRNAs. I have only a few suggestions.

Point 1: The authors state that “only 54 DEmiRNAs have successfully predicted target genes”.  But this is a bit worrying. Is this normal for this type of study? If these are real miRNAs why is the percentage of putative targets so low?

Respond 1: Thanks to the reviewer for your advice. We used two software, miRanda and Targetscan, to predict miRNA targets, and those target genes predicted by both the two software were regarded as the final targets. So, because of the restriction of this strict condition, some DEmiRNAs were not predicted target genes, but in fact, they were predicted targets by one of the two software. Moreover, the percentage of putative miRNA targets are also low in other studies, please see the following two references.

Zhao, X.; Yang, G.; Liu, X.; Yu, Z.; Peng, S., Integrated analysis of seed microRNA and mRNA transcriptome reveals important functional genes and microRNA-targets in the process of walnut (Juglans regia) seed oil accumulation. Int. J. Mol. Sci. 2020, 21, (23), 9093. Doi: 10.3390/ijms21239093.

Gao, Y.; Wu, F.; Ren, Y.; Zhou, Z.; Chen, N.; Huang, Y.; Lei, C.; Chen, H.; Dang, R., MiRNAs expression profiling of bovine (Bos taurus) testes and effect of bta-mir-146b on proliferation and apoptosis in bovine male germline stem cells. Int. J. Mol. Sci. 2020, 21, (11), 3846. Doi: 10.3390/ijms21113846.

Point 2: The suggestion of finding 135 new miRNAs is intriguing. Do these ‘new’ miRNAs show any homology with miRNAs from more distantly related taxa at all?

Respond 2: Thanks to the reviewer for your advice. Yes, many new miRNAs are homologous to miRNAs from more distantly related taxa. We added a new Table S2 to show the information.

Point 3: Can the authors provide a bit more information on exactly where in the genome these miRNAs are being transcribed? That is, where are these miRNA genes located? Which chromosomes? Are the novel miRNAs being expressed from certain clusters? I’d be curious to know more information.

Respond 3: Thanks to the reviewer for your advice. The position of miRNAs was obtained by aligning the miRNAs precursor sequences with the A. mellifera reference genome. We added the information in Table S2. The novel miRNAs are not expressed from certain clusters.

Point 4: I was impressed that the research team did some analyses to see if the putative miRNA actually targeted a particular mRNA. They seem to have found results suggesting that it did. I would have welcomed more directed functional assays.

Respond 4: Thanks to the reviewer for your advice. In this study, we mainly analyzed the expression profile of miRNAs in hypopharyngeal gland of honey bees, and obtained many valuable cues for further study. Therefore, in the subsequent studies, we will study the specific functions of these miRNAs in the development of hypopharyngeal gland, especially novel-miR-11 and its two target genes.

Reviewer 2 Report

The main aim of this research was adaptation of the small RNA sequencing to identify DEmiRNAs between three developmental stages of A. mellifera hypopharyngeal gland.

The methodology used in this work is properly applied and well described in manuscript.

The results are relevant for better understanding of regulating of hypopharyngeal gland development and royal jelly secretion.

The conclusions are consistent with the results. Tables and figures are correct.

However, some important points have to be addressed and details need to be added:

Line 12: Remove abbreviation from Abstract.

Line 16 and line 18: same as previous.

Line 28: “large amount of royal jelly proteins” be more precise, what does mean large amount?

Lines 37-38: hormones or pheromones, please add references for this statement.

Line 54 and 61: Explain abbreviations here, having in mind they will be removed from abstract.

Line 70; Latin names should be written in italic.

Line 225: “similar population” – please explain, similar in what, number, genetics, strength etc, and add the method of estimation. Add details about genetics and strength od colonies.

Line 226: is 80% humidity too high, please add a reference for this.

Line 232: were glands kept dry or in some medium/shield?

Author Response

Dear editor Codruta Cormos,

We greatly appreciate you and two reviewers giving so many very helpful and useful comments for our manuscript “Expression profile of microRNAs during development of the hypopharyngeal gland in honey bee, Apis mellifera” (ijms-1950892). These comments greatly improve the quality of our manuscript.

We have carefully considered each point raised in the reviewers, and provide point-by-point responses to the reviewers’ comments below. The corrections are highlighted in red in the revised version. We hope we have addressed all comments, and please let us know if more information is needed. Thank you once again for considering our manuscript.

Best wishes,

Prof. Dr. Zilong Wang

Director of Honeybee Research Institute

Jiangxi Agricultural University

Nanchang, Jiangxi, 330045

  1. R. of China

Response to Reviewer 2 Comments
Comments:

The main aim of this research was adaptation of the small RNA sequencing to identify DEmiRNAs between three developmental stages of A. mellifera hypopharyngeal gland.

The methodology used in this work is properly applied and well described in manuscript.

The results are relevant for better understanding of regulating of hypopharyngeal gland development and royal jelly secretion.

The conclusions are consistent with the results. Tables and figures are correct.

However, some important points have to be addressed and details need to be added:

Point 1: Line 12: Remove abbreviation from Abstract.

Respond 1: Thanks to the reviewer for your advice, we have revised it according to your suggestion.

Point 2: Line 16 and line 18: same as previous.

Respond 1: Thanks to the reviewer for your advice, we have revised it according to your suggestion.

Point 3: Line 28: “large amount of royal jelly proteins” be more precise, what does mean large amount?

Respond 3: According to Li et al., 150 queen cell cups with grafted larvae were put into Apis mellifera colonies, and they collected 35.28±0.91g royal jelly after 72h, even collected 151.92±1.49g royal jelly in high royal jelly producing honeybee colonies. Thus, the bees can produce a large amount of royal jelly during the nursing stage. However, there is no report of the amount of royal jelly produced by single nurse bee, so, we deleted the “a large amount of”.

Li, J.; Feng, M.; Begna, D.; Fang, Y.; Zheng, A., Proteome comparison of hypopharyngeal gland development between Italian and royal jelly producing worker honeybees (Apis mellifera L.). J. Proteome Res. 2010, 9, (12), 6578-6594. Doi: 10.1021/pr100768t.

Point 4: Lines 37-38: hormones or pheromones, please add references for this statement.

Respond 4: Thanks to the reviewer for your advice, we have revised it according to your suggestion and added references in lines 37-38.

Point 5: Line 54 and 61: Explain abbreviations here, having in mind they will be removed from abstract.

Respond 5: Thanks to the reviewer for your advice, we have revised it in lines 35, 58 and 66 according to your suggestion.

Point 6: Line 70: Latin names should be written in italic.

Respond 6: Thanks to the reviewer for your advice, we have revised it according to your suggestion.

Point 7: Line 225: “similar population” – please explain, similar in what, number, genetics, strength etc, and add the method of estimation. Add details about genetics and strength od colonies.

Respond 7: Here, similar population means that each colony has the same number of combs and similar number of bees, larvae and caped brood. We estimated the approximate number of bees mainly by observation.

Point 8: Line 226: is 80% humidity too high, please add a reference for this.

Respond 8: Thanks to the reviewer for your advice, we have revised it in line 243 according to your suggestion.

Point 9: Line 232: were glands kept dry or in some medium/shield?

Respond 9: Thanks to the reviewer for your advice, we have revised it. The samples were ground into powder and mixed with TransZol (TransGen Biotech, China) in plastic Eppendorf tubes, then all the samples were promptly frozen and stored in liquid nitrogen until RNA extraction.
